# Contribution of Epigenetic Mechanisms in the Regulation of Environmentally-Induced Polyphenism in Insects

**DOI:** 10.3390/insects12070649

**Published:** 2021-07-15

**Authors:** Gautier Richard, Julie Jaquiéry, Gaël Le Trionnaire

**Affiliations:** INRAE Bretagne-Normandie-UMR 1349 IGEPP, Domaine de La motte BP 35327, CEDEX, 35653 Le Rheu, France; gautier.richard@inrae.fr (G.R.); julie.jaquiery@inrae.fr (J.J.)

**Keywords:** seasonal, dispersal and caste polyphenism, chromatin, DNA methylation, histone post-translational modifications, non-coding RNAs

## Abstract

**Simple Summary:**

Polyphenism is a widespread phenomenon in insects that allows organisms to produce alternative and discrete phenotypes in response to environmental conditions. Epigenetic mechanisms, including histone post-translational modifications, DNA methylation and non-coding RNAs, are essential mechanisms that can promote rapid and flexible changes in the expression of transcriptional programs associated with the production of alternative phenotypes. This review summarizes knowledge regarding the contribution of those mechanisms in the regulation of the most-studied examples of polyphenism in insects.

**Abstract:**

Many insect species display a remarkable ability to produce discrete phenotypes in response to changes in environmental conditions. Such phenotypic plasticity is referred to as polyphenism. Seasonal, dispersal and caste polyphenisms correspond to the most-studied examples that are environmentally-induced in insects. Cues that induce such dramatic phenotypic changes are very diverse, ranging from seasonal cues, habitat quality changes or differential larval nutrition. Once these signals are perceived, they are transduced by the neuroendocrine system towards their target tissues where gene expression reprogramming underlying phenotypic changes occur. Epigenetic mechanisms are key regulators that allow for genome expression plasticity associated with such developmental switches. These mechanisms include DNA methylation, chromatin remodelling and histone post-transcriptional modifications (PTMs) as well as non-coding RNAs and have been studied to various extents in insect polyphenism. Differential patterns of DNA methylation between phenotypes are usually correlated with changes in gene expression and alternative splicing events, especially in the cases of dispersal and caste polyphenism. Combinatorial patterns of histone PTMs provide phenotype-specific epigenomic landscape associated with the expression of specific transcriptional programs, as revealed during caste determination in honeybees and ants. Alternative phenotypes are also usually associated with specific non-coding RNA profiles. This review will provide a summary of the current knowledge of the epigenetic changes associated with polyphenism in insects and highlights the potential for these mechanisms to be key regulators of developmental transitions triggered by environmental cues.

## 1. Introduction

Organisms have to cope with the constant fluctuations of their biotic or abiotic environment. When these changes are perceived and integrated, the organism can potentially adjust its phenotype by modifying its physiology accordingly to enhance its ability to face these new environmental conditions. The ability for a genotype to produce different phenotypes in response to fluctuations of external cues is defined as phenotypic plasticity [1]. This plasticity relies on the capacity of the genome to express different combinations of genes and proteins as a response to external signals in order to produce an adjusted phenotype. Phenotypic plasticity can be continuous, with gradual phenotypes being produced depending on the environment, ranging from subtle metabolic adjustments to wider morphological changes at the scale of tissues or even organs [2]. Phenotypic plasticity can also result in the production of discrete phenotypes, often exhibiting extreme morphological, physiological and behavioural differences. Such a discrete phenotypic plasticity is defined as polyphenism [3]. Polyphenisms are widespread in insects [4]. The existence of very discrete developmental stages throughout the life cycle of many insects is the most obvious example. Larval, pupal and adult stages indeed display very different morphological and ecological traits in holometabolous insects (e.g., Coleopteran, Lepidoptera, Hymenoptera, Diptera) thus highlighting the ability of a unique genome to express different gene and protein combinations at various steps of the life cycle. This genome expression plasticity is not triggered by external cues but by internal factors. Remarkably, insects also display other types of polyphenism that are induced by external stimuli. These are referred to as environmentally-induced polyphenism and provide insects with an extraordinary ability to rapidly adapt their phenotype to a wide array of environmental conditions. The signals triggering these environmental changes can be abiotic (e.g., temperature, photoperiod), as is the case for seasonal polyphenism, where insects produce discrete morphs adapted to seasonal constraints (e.g., wing patterning in butterfly or reproductive mode alternation in aphids). However, these signals can also be biotic. For example, dispersal polyphenism relies on the production of alternative morphs that are morphologically and behaviourally adapted to dispersion (e.g., phase polyphenism in locusts and wing polyphenism in aphids). In those cases, an increase in population density and subsequently of tactile stimuli [5] as well as an impoverishment or rarefaction of nutritional resources [6] will trigger the development of alternative phenotypes. In social insects (ants, bees, wasps and termites) where reproductive and non-reproductive tasks are partitioned within the colony (caste polyphenism), the food given to the larva generally determines if it will develop into the worker or queen caste. Although these three types of polyphenisms (i.e., seasonal, dispersal and caste polyphenisms) are induced by different environmental signals, they share common features in their regulatory scheme (Figure 1). External cues first need to be perceived and integrated by specific groups of cells, tissues or organs that are dependent on the nature of the signal itself. This processed signal must then be transduced towards the target tissues/organs that will express alternative genetic programs (summarized in Table 1). In seasonal, dispersal and caste polyphenisms, the neuro-endocrine system is a central integrator and transducer of external cues [7]. The nervous structures, neuropeptides or hormones involved vary between species and polyphenisms, but are well documented [4]. Finally, once the target tissues/organs receive the signal, the genome of the corresponding cells will modify its expression allowing the transcription of new sets of genes and proteins necessary for the production of the alternative phenotype. Such genome-wide expression modifications rely on mechanisms that allow for rapid, flexible changes in transcriptional status of many and distant genes in a coordinated manner, such as epigenetic mechanisms.

Epigenetic mechanisms are major regulators of gene expression in all organisms, including insects [8], and are thus involved in the production of alternative phenotypes. Epigenetics encompasses all the processes of phenotypic variation that are not due to DNA sequence variation. These phenotypic changes might also be heritable across generations. Epigenetic mechanisms involved in the production of alternative phenotypes may occur at different levels. A first level involves the structure of the chromatin. In the nucleus, the DNA containing the coding information is wrapped around histone octamers to form the nucleosomes that correspond to the basic unit of chromatin (Figure 1). Nucleosome distribution and density all along the DNA sequence can affect chromatin accessibility to transcriptional machinery and consequently gene expression levels. Nucleosome stability and distribution is notably controlled by histone post-translational modifications (PTMs). Chromatin remodelling triggered by histone PTMs thus corresponds to a first level of possible epigenetic modifications [9]. DNA methylation is a second major epigenetic mechanism that can affect gene expression levels. DNA methylation involves the transfer of a methyl group onto the C5 position of the cytosine to form 5-methylcytosine (5mC). Changes in DNA methylation levels are associated with gene expression activation or silencing, depending on the organism and the genomic context [10], and have been shown to play key roles in insects (refer to references in [8]). Other changes of the DNA include 5-hydroxymethylcytosine (5hmC), 5-formylcytosine (5fC), 5-carboxylcytosine (5caC) and N ^6^-methyladenine (6mA), and play critical roles in many biological processes [11,12,13,14,15,16,17,18,19]. A third level for the epigenetic control of alternative phenotypes involves small and long non-coding RNAs (sncRNAs and lncRNAs, respectively). Among sncRNA, miRNAs (for microRNAs) and siRNAs (for small interfering RNAs) are major post-transcriptional gene expression regulators. PiRNAs (for piwi-interacting RNA) and siRNAs can also act as intermediates to direct DNA methylation or histone PTMs at specific genome locations, for example by controlling transposable element expression in germline cells in the case of piRNAs [20]. LncRNAs also share the potential for regulating gene expression at both transcriptional and post-transcriptional levels [21]. Finally, chromatin tridimensional organisation orchestrates gene transcription by regulating chromosome-wide (chromosome territories, A/B compartments) as well as local chromatin topology structures (loops, topologically associating domains) supporting transcription binding factors, such as enhancers or insulators [22,23]. Altogether, the combination of these three major epigenetic mechanisms can have a strong impact on the coordinated regulation of gene expression in the context of alternative phenotype production associated with polyphenism in insects.

In the next sections, we will review the contribution of chromatin regulations, i.e., histone PTMs, DNA methylation and non-coding RNAs in the regulation of insect polyphenisms and their associated transcriptomic programs (summarized in Table 1). Various categories of insect polyphenisms are documented in the literature; however, molecular data detailing the involvement of epigenetic mechanisms can be missing. Moreover, specific phenotypic changes can sometimes be assigned to distinct polyphenisms, for instance, dispersal polyphenism in aphids, which is also a defensive polyphenism when induced by predators [24]. Polyphenism classification can thus be somewhat unclear. In this review, we focus on environmentally-induced polyphenisms in well-studied insect models, classified as three different groups: seasonal, dispersal and caste polyphenisms. For each polyphenism, we start with a brief description of the neuro-endocrine transduction of the triggering signals and follow with a picture of the transcriptional and epigenetic modifications that contribute to the production of alternative phenotypes.

## 2. Seasonal Polyphenism

Photoperiod and temperature are the most predictable environmental cues organisms can perceive and integrate to anticipate the arrival of a new season. Some species of insects display a seasonal polyphenism, with discrete morphs each adapted to the different seasonal constraints. The two most studied examples so far are wing patterning polyphenism in butterflies as a response to temperature and the reproductive polyphenism in aphids triggered by autumnal photoperiod shortening.

### 2.1. Wing Patterning Polyphenism in Butterflies

Many butterfly species exhibit a wing patterning polyphenism. The southern African butterfly, *Bicyclus anynana*, is the most studied example to date. During the wet season, butterflies exhibit prominent marginal eyespots on the surface of their hind wings while in the dry season these eyespots are reduced (Figure 1). In addition, dry season morphs are less active and brownish in colour, probably to better hide from predators in the dry vegetation [25]. The temperature experienced by larvae in their final stage of development is the main external cue governing this polyphenism, with high temperatures (27 °C) leading to the production of the wet season morph, and lower temperatures (20 °C) leading to the production of the dry season morph. Such a phenotypic switch is triggered by the timing of the hormonal peak of circulating ecdysteroids in the haemolymph. At low temperatures, the ecdysteroid peak occurs later during pupal development, while it occurs earlier at higher temperatures [26]. Little is known about the transcriptomic and epigenetic events related to this polyphenism, apart from the involvement of the *distal-less* gene in the stipulation of a focal region of cells on the wing, allowing the formation of the eyespot [27] and possibly the regulation by ecdysteroid levels [28].

### 2.2. Reproductive Polyphenism in Aphids

Aphids are hemipterous insects characterized by complex life cycles on diverse and multiple host plants, including many crops. They display major polyphenic traits during their life cycle, including a dispersal polyphenism, allowing them to potentially escape stressful environments (see below) and a seasonal polyphenism resulting in the production of discrete morphs with distinct reproductive modes. Changes in day length, i.e., photoperiod, are the main trigger of this reproductive mode switch [29,30]. Indeed, during spring and summer (under long photoperiod conditions), aphids reproduce exclusively by viviparous parthenogenesis. In their embryos, only diploid oocytes are produced through apomictic parthenogenesis, ending up with genetically identical offspring. Parthenogenesis thus allows aphid populations to colonize rapidly new habitats (notably crops). At the end of the summer, parthenogenetic females perceive the photoperiod shortening and transduce this predictable signal towards their embryos that change their reproductive fate, ending up with the production of sexual morphs (oviparous females and males) that display true haploid gametogenesis. Hence, the shortening of the photoperiod in aphids induces the production of sexual morphs. The first step of this process consists in the photoperiod signal perception, followed by neuro-endocrine transduction. Physiological studies first revealed that a group of neurosecretory cells (called “Neurosecretory Cells Group I”) within the brain were involved in the reception of the decreasing photoperiodic signal, as micro-cauterization of those cells abolishes the photoperiod response [31]. Pharmacological studies then showed that juvenile hormone was playing an essential role in the neuro-endocrine transduction of the signal [32]. Several transcriptomic studies (genome-wide and candidate gene approaches) identified significant gene expression changes between long and short photoperiod conditions in the heads of aphids. Although the causality of these transcriptomic changes remains unknown, these studies highlight the variety of signalling pathways that might be involved in this process. Indeed, it has been suggested that opsin could be involved in photoreception [33,34] and that the circadian clock might be a component of the photoperiodic clock [35,36]. Additional studies showed that neuropeptides, such as insulin [37], as well as neurotransmitters, such as dopamine [38,39,40], could be involved in the neuro-endocrine transduction of the photoperiodic signal. Nevertheless, little is known so far about the epigenetic modifications that trigger these early transcriptomic changes. The annotation of the gene repertoire of DNA methylation [41] and chromatin remodelling [42] systems nevertheless suggests that these are functional in aphids. A genome-wide comparison of the transcriptome of embryos containing haploid germlines (corresponding to parthenogenetic females) and diploid germlines (corresponding to oviparous females) revealed differentially expressed gene coding for histone proteins (H1 and H2B.3), a histone methylase (*Suv4–20H1*) and *uhrf1*, a protein known to bind histone residues. This suggests that chromatin remodelling events might be key triggers of embryonic germline fate switches [43]. The contribution of epigenetic mechanisms during the embryo fate switch has not been investigated yet. Nevertheless, two recent studies aimed at comparing the epigenome of adult parthenogenetic and sexual morphs of aphids. Richard et al. (2017) compared the open chromatin and transcriptomic profiles of parthenogenetic females and males in the pea aphid [44]. They found that the chromatin of the single X chromosome of males was more accessible than the chromatin of the two Xs of females, probably to compensate for the missing second X chromosome transcriptional dose in the male genome. This was positively correlated with an enhanced transcription of thousands of genes on the X chromosome involving male-specific functions. Altogether, these data suggest the existence of a chromosome-wide epigenetic regulation of gene expression in males for dosage compensation as well as for the establishment of male-specific transcriptomic programs. Another study recently compared the genome-wide DNA methylation pattern of parthenogenetic females and males in *Myzus persicae* [45]. They showed that X chromosomes and autosomes had very different methylome profiles between sexes; autosomal genes are hypomethylated in males compared with females, while X-linked genes are hypermethylated in males. Further analyses are now required to understand the contribution of these epigenetic mechanisms at the onset of the transition between parthenogenetic and sexual reproduction as a direct response to photoperiod shortening signal.

## 3. Dispersal Polyphenism

When environmental conditions are locally favourable, insect individuals invest preferentially in reproduction rather than dispersal. However, following the degradation of habitat quality, the production of phenotypes suitable for dispersion may be favoured. Dispersal polyphenism is such a possibility and has been described in several insect groups [46]. Phase polyphenism in locusts is one of the most studied examples, whereby a switch from a solitary to a gregarious (and migratory) phenotype occurs in response to increasing individual densities. In aphids, wingless parthenogenetic females are able to produce winged parthenogenetic females in their progeny when exposed to various signals indicative of a deterioration of habitat, including the presence of predators and food quality decrease.

### 3.1. Phase Polyphenism in Locusts

Locusts are responsible for major economic losses when they form migratory swarms that devastate large crop surfaces in a very limited time. Locusts display solitary and gregarious morphs (known as phase polyphenism), which differ in morphology (body size and colour, Figure 1), neurochemistry, physiology and behaviour [47]. When exposed to crowding, individuals with an initial solitary phenotype can initiate a gradual transition towards a gregarious behavioural phenotype in just a few hours, as behavioural changes rely on short-term neuronal plasticity. In contrast, the development of morphological changes associated with the gregarious phenotype extend over several generations [47]. Two distinct sensory pathways are involved in the phase transition in the desert locust (*Schistocerca gregaria*): the cerebral pathway, involving visual and olfactory stimuli, and the thoracic pathway, induced by tactile information [48]. Once the crowding signals are perceived and integrated, the neuroendocrine system will initiate the phase transition. Serotonin is a key neurotransmitter involved in the control of phase transition. Indeed, its hemolymph concentration dramatically increases within hours after crowding in the desert locust [49]. Such effect is probably mediated through protein kinase A signalling [50]. Dopamine is also suggested in *Locusta migratoria* to act as a gregarization factor [51] while juvenile hormone and corazonin are likely to be involved in body colour differences between the solitary and gregarious morphs [52], which harbour brown or yellow colour, respectively (Figure 1). Transcriptomics approaches were also conducted to identify genes differentially expressed between solitary and gregarious phenotypes. First, EST datasets from the migratory locust revealed a repression of anabolic, biosynthetic and muscle-specific gene expression in the hindleg, midgut and head tissues of gregarious versus solitary locusts [53]. A strong upregulation of a subset of juvenile hormone binding proteins was also observed in the head of gregarious locusts, thus providing a link between transcriptional changes and hormonal signalling of crowding. A comparison in the desert locust of central nervous system (CNS) transcriptome during the two phases revealed the upregulation of heat-shock proteins (HSP) and immunity genes in gregarious locusts [54], suggesting that population density increase might induce a general physiological stress state. A reduction of energy metabolism and biosynthesis was also observed in gregarious locusts in another transcriptomic study in the migratory locust [55], together with the overexpression of gene coding for synaptic transport components, neuromodulators receptors and neurotransmitter synthetases, confirming the neuronal modulation of phase transition. A microarray study on fourth instar migratory locusts during phase transition then revealed the differential expression of chemosensory (CSP) and *takeout* proteins, corresponding to large families of soluble binding proteins secreted in the haemolymph [56]. Gregarious and solitary phenotypes are thus associated with profound transcriptomic modifications even if the causality of these changes remains unknown.

As major regulators of rapid changes in gene expression, epigenetic mechanisms are likely to play an important role in phase transition in a tissue-specific manner. So far, DNA methylation is the most studied epigenetic mechanism in the context of phase polyphenism in locusts. It was first demonstrated that the *Schistocerca gregaria* genome is relatively highly methylated (1.3–1.4%) compared with other insects (less than 1% in drosophila [57]). More precisely, 5mC (i.e., methylated cytosine) are found in both coding regions and repetitive elements [58]. Such trends were confirmed in the *Locusta migratoria* genome [59]. A methylome analysis of brain tissues from fourth instar solitary and gregarious *L. migratoria* locusts first revealed that methylation in coding regions was lower and more variable overall than in the rest of the genome. Intergenic repetitive elements were especially highly and stably methylated, and introns were more methylated than exons [60]. 90 genes were then identified as differentially methylated between the two phenotypes, including genes coding for signal transduction proteins and some others involved in microtubule cytoskeleton organization, and thus hypothesized to contribute to synaptic plasticity, since these cellular components are known to play a crucial role in neuronal functions in animals. The functional role of DNA methylation in the control of phase transition is nevertheless not demonstrated yet. Regarding the precise role of chromatin remodelling, little is known on their exact role in phase polyphenism. Histone marks may play a role since H3 histones are more phosphorylated in brains of gregarious than of solitary individuals [47]. Small RNAs transcriptome profiling during the two phases allowed for the identification of 185 miRNAs family candidates, thousands of endogenous siRNAs, some of them of transposon origin, as well as many piRNAs [61]. Interestingly, comparison of small RNA expression patterns of the two phases revealed that longer small RNAs are expressed more abundantly in the solitary individuals, suggesting a specific role in transcript level regulation during this phase. Taken together, the precise role of epigenetic mechanisms in the regulation of phase polyphenism requires further investigation and probably at a tissue-specific level, considering the variety of neuronal and morphological modifications that are at stake during this process.

### 3.2. Wing Polyphenism in Aphids

During the parthenogenetic part of their life cycle, aphids display a dispersal polyphenism when wingless females produce winged morphs by parthenogenesis in response to various environmental cues, including population density (or crowding), the physiological state of the plants they feed on and the presence of predators or parasites [62]. Winged and wingless parthenogenetic females differ in several morphological, physiological and behavioural characteristics, notably the presence or absence of a functional flight apparatus as well as differences in the level of cuticle sclerotization in the head and thorax [63]. Once crowding signals are perceived, the neuro-endocrine system plays an essential role in its transduction, juvenile hormone being shown to be implicated [6]. A recent transcriptomic comparison of parthenogenetic female heads of the pea aphid submitted or not to a crowding treatment revealed profound differences in gene expression associated with the maternal signalling inducing the winged morphs [64]. Some significant transcriptional changes were found for genes involved in odorant binding, neurotransmitter transport and hormonal signalling. Interestingly, genes from the dopamine and ecdysone pathway were found as differentially expressed, suggesting a role for these molecules in maternal signalling too. The role of ecdysone was then more precisely investigated. Indeed, pharmacological approaches (ecdysone feeding, ecdysone analog and ecdysone receptor antagonist injections) combined with RNAi directed against Ecdysone Receptor (EcR) functionally validated the role of ecdysone signalling in wing polyphenism: an increase in ecdysone signalling is associated with a higher production of wingless individuals, while a decrease in ecdysone signalling promotes the production of winged progeny [65]. Regarding the contribution of epigenetic mechanisms in wing polyphenism, little is known so far apart from a handful of genes involved in chromatin remodelling that are differentially expressed in the heads of crowded and uncrowded parthenogenetic females. This suggests a role for this mechanism in the maternal signalling of wing polyphenism [64]. A more recent study investigated the extent of alternative gene splicing between winged and wingless parthenogenetic females of the pea aphid, at both embryo and adult stages [66]. Overall, they found that 28% and 27% of the expressed genes displayed alternative splicing in winged and wingless adult females, respectively, and 25% for winged-destined and unwinged-destined embryos. When comparing these phenotypes, 332 differentially spliced sites were identified between winged and wingless adults but only 13 between winged and unwinged-destined embryos. The majority of differentially spliced genes were nevertheless not differentially expressed. This study suggests a potential role for alternative splicing in the expression of environmentally induced phenotypes. The causality of these differences is not yet known, but DNA methylation might be a good candidate to explain these splicing patterns (see below). Altogether, epigenetic data are so far very limited in the case of dispersal polyphenism and further investigations are required.

## 4. Caste Polyphenism

Eusocial insects (i.e., bees, wasps, ants and some termites) display a highly sophisticated organization within the colony, with the presence of different groups of individuals that differ in their morphology and behaviour, which are referred to as caste. The reproductive caste includes queens and males, and the non-reproductive caste includes workers (and soldiers in some species). Caste morphology is generally determined early during larvae development in response to environmental cues (see Lavanchy and Schwander, 2019 [67]). Interestingly, the worker caste is in charge of various tasks, including nursing, foraging, nest maintenance, defence or policing [68]. Within the non-reproductive caste, behaviour can sometimes change in an age-specific manner, such as the nursing to foraging transition in honeybees [69]. In other species (e.g., some ants and termites), non-reproducing individuals can differ in morphology, the largest one being specialized in defence tasks (soldiers or major workers) and the small ones (minor workers) in other tasks. In this section, we will mainly focus on the regulation of caste polyphenism in honeybees and ants. Unlike seasonal and dispersal polyphenisms, the contribution of epigenetic mechanisms in the regulation of caste polyphenism has been well documented in the last few years, especially DNA methylation, chromatin regulation and transcription factors. The contribution of non-coding RNAs is also being considered.

### 4.1. Nutrient Sensing and Neuro-Endocrine Signalling as Key Triggers of Caste Differentiation

Depending on the environmental stimuli perceived at larval or nymphal stages, individuals can be routed towards alternative developmental programs and produce distinct phenotypes once they are adults. They are especially able to produce either reproductive (queen) or non-reproductive (workers) phenotypes. In ants, the amount and composition (including specific chemical compounds) of larval diet is a widespread stimulus that triggers caste fate determination. In *Myrmica rubra*, a highly nutritive diet during larval growth, can induce queen development while starvation can prevent it [70,71]. However, other signals, such as temperature or pheromone, can also influence developmental trajectories [72]. Internal factors, such as genetic variations, can also play a role in caste determination; polymorphism in sequences that regulate gene expression (Transcription factors binding sites or ncRNAs) can be used to modulate thresholds that define an individual’s developmental trajectory [73]. More intriguing is the case of social hybridogenesis in some ant species, an unusual form of reproduction found in hybrids between different species. Indeed, queens are produced by parthenogenesis or from intra-lineage mating while workers are only produced from hybrid crosses between genetically divergent lineages [67]. Nevertheless, little is known about the neuro-endocrine transduction of this nutritional signal towards the target tissues apart from a recent study demonstrating that the neuropeptide corazonin was involved in the control of social behaviour as well as caste identity in ants [74].

In honeybees, larvae fed with royal jelly differentiate into queen, while larvae that do not receive this diet develop as workers. Interestingly, it has been shown that royalactin, a 57 kDa protein present in royal jelly, is necessary to induce the differentiation of honeybee larvae into queens [75]. This protein promotes the increase in body size and the development of ovaries. More precisely, royalactin activates the Epidermal growth factor receptor (EGRF), which then turns on the InR substrate (IRS), a well-known target of insulin/insulin-like growth factor (IGF) signalling pathway, but also TOR nutrient-signalling pathway. Juvenile hormone signalling also appears to play an essential role in honeybee caste determination, especially by promoting ovarian development [76]. Diet sensed by larvae are thus transduced by the neuro-endocrine system to promote distinct caste fates later during development. Epigenetic mechanisms then act as key regulators in targeted cell types to establish alternative developmental programs.

### 4.2. DNA Methylation Patterns Changes during Caste Polyphenism

Eusocial insect genomes, including bees, ants, some wasps and termites, harbour the two key DNA methyltransferases, DNMT1-responsible for maintenance of DNA methylation and DNMT3, involved in *de novo* DNA methylation [68]. In contrast, at least one or both genes are missing in non-social insects, such as *Drosophila*, silkworm and red flour beetle. These findings raised a strong interest for the functional role of this epigenetic mechanism in the regulation of caste polyphenism. Kucharski et al. (2008) first showed that RNAi silencing of DNMT3 in honeybee worker larvae resulted in the production of queen-like ovaries, which suggested that DNA methylation might be necessary for inhibiting queen development [77]. Elango et al. further demonstrated that the *A. mellifera* genome displays a bimodal distribution of CpG content across genes sequences (also found in *A. pisum* [78]), with functionally distinct low and high CpG methylated genes; the latter displaying a strong bias towards caste-specificity expression [79]. Following these observations, DNA methylation patterns in the brain of adult honeybee workers and queens were compared [80]. Nearly all methylated cytosines were located in gene exons. Genes (#550) were identified as differentially methylated between castes, thus revealing a potential role for DNA methylation in the regulation of gene expression. A strong correlation between methylation patterns and splicing sites was also found, suggesting a role for DNA methylation in alternative splicing. Later on, DNA methylation patterns were compared between larval and adult stages in honeybee workers and queens [81]. This analysis revealed that larval stages displayed five times more differentially methylated genes compared to adult stages, and that a significant proportion of these genes were upregulated in worker larvae compared to queen larvae. Among those genes, several are involved in metabolic and signalling pathways, including juvenile hormone and insulin, two hormones/peptides known to regulate caste determination. Strong correlations between DNA methylation and alternative gene splicing were also found in this study, and particularly for the *alk* gene, a receptor tyrosine kinase acting as an important regulator of metabolism, thus linking diet experienced by the larvae and caste determination. RNAi knockdown of DNMT3 also induced some changes in gene splicing patterns, thus confirming the role of DNA methylation in the regulation of alternative splicing [82].

In ants, whole-body DNA methylation patterns were examined in two species (*Camponotus floridanus* and *Harpegnathos saltator*), confirming that methylated cysotines were strongly enriched in the exons of active genes. As seen in bees, changes in exonic DNA also correlated with alternative splicing, such as exon skipping. Overall, few changes in methylation levels were observed between castes in both ant species. Interestingly, several genes conserved between the two species displayed caste-specific changes in DNA methylation, especially genes involved in reproduction, telomere maintenance and non-coding RNA metabolism [83]. It was also reported that the termite *Zootermopsis nevadensis* displayed some of the highest levels of DNA methylation found in insects: 12% of the genomic CpGs dinucleotides and 58% of the exonic CpGs [84]. Interestingly, strong differences in methylation between castes were observed. Differentially methylated genes were also more frequently alternatively spliced, and preferentially involved in developmental-related functions. In addition, intergenic differentially methylated regions were enriched for multiple transcription factor binding sites, suggesting an interplay between DNA methylation and transcriptional regulation of gene expression [84]. Altogether, it appears that the global trends related to the role of DNA methylation in the regulation of caste polyphenism are shared between bees, ants and possibly termites.

### 4.3. Contribution of Transcription Factors and Chromatin Remodelling Events in Caste Polyphenism

In bees, a few indirect cues first suggested a role of chromatin regulation and histone PTMs in the control of caste polyphenism. Mass spectrometry approaches showed that histone H3.1, H3.3 and H4 were extensively modified by lysine methylation and acetylation in honeybees [85]. The authors then compared histone PTMs within queen ovaries and 96-h-old whole larvae. They observed similar global profiles of histone PTMs in both samples as well as some specific combinatorial patterns of lysine methylation on H3K27 and H3K36 residues more frequently identified in histones extracted from queen ovaries than from larvae. Interestingly, the comparison of brain methylomes from queen and worker castes [80] allowed for the identification of the differential methylation of specific histone genes. They found that only intron-containing histone variants were methylated, whereas intronless canonical histone genes were not. This suggests that the interplay between DNA methylation and histone profiles might be important for modulating chromatin accessibility and gene expression in the context of caste polyphenism. Interestingly, it has been shown that royal jelly, known to induce the queen phenotype, has histone deacetylase inhibitor activity (HDACi). Indeed, the fatty acid (E)-10-hydroxy-2-decenoic acid (10HDA), accounts for up to 5% of royal jelly and harbours this HDACi activity, which suggests that the inhibition of histone deacetylation is important for queen caste development [86]. More recently, a genome-wide comparison of various histone PTMs between queen-destined and worker-destined larvae highlighted extensive differences in H3K4me3, H3K27ac as well as H3K36me3 [87]. Many of these were positively correlated with caste-specific gene expression. Intronic H3K27 acetylation was also found to be a trigger of worker caste differentiation. This study thus demonstrates the role of chromatin modifications in the establishment and maintenance of caste-specific transcriptional programs at an early stage of larval development in the honeybee.

In ants, regulatory elements and genome-wide histone PTMs related to caste polyphenism also received strong attention [88]. First, genome sequence comparisons of eight eusocial (one honeybee and seven ant species) and 22 solitary insects (spanning different insect orders) revealed that the transcription factor binding sites (TFBS) within the promoters of orthologous genes are more divergent among eusocial insects than between solitary and eusocial insects [89]. Genes displaying the most important evolutionary changes are involved in neuroendocrine signalling and are differentially expressed between castes in *Camponotus floridanus* and *Harpegnathos saltator* species. These genes contain some binding motifs for neuronal-related TFs in their promoters, including cyclic AMP response element binding protein (CREB), empty spiracles (EMS) and grainyhead (GRH), suggesting that neuronal gene networks have been the target of regulatory rewiring in the course of evolution. In *Camponotus floridanus,* genome-wide comparison of various histone PTMs between two female workers (major and minor) and male phenotypes showed that H3K27ac discriminates between these different castes, but also explains most of the differential gene expression between castes [90]. The genes showing these correlated changes in gene expression and H3K27ac patterns are involved in muscle development and neuronal regulation as well as sensory response. Interestingly, binding sites for CBP, which is the transcriptional coactivator CREB binding protein, show important variations in H3K27ac patterns between castes. Considering the role of CBP as a major acetyltransferase of histone H3 lysine 27 residues in insects [91], this suggests that gene expression changes associated with caste polyphenism establishment might be mediated by differential recruitment of CBP to chromatin.

### 4.4. Non-Coding RNAs and Caste Polyphenism

The role of small RNAs and long non-coding RNAs in the regulation of gene expression during caste polyphenism has been investigated mainly in ants and honeybees. In particular, miRNAs are well-known post-transcriptional regulators of gene expression. In ants, respectively 96 and 159 miRNA genes were annotated within *Camponotus floridanus* and *Harpegnathos saltator* genomes. Interestingly, the minor and major *C. floridanus* worker castes displayed differential expression of miRNAs; mir-64 being, for example, upregulated in minor worker castes [92]. Later on, small RNA profiling allowed for the identification of 20 novel miRNAs in *C. floridanus*, 12 of them displaying stage- and caste-specific expression [89]. Small RNA profiling in honeybees unveiled the differential expression of specific miRNA genes in the brain associated with age-dependent behavioural changes [93] as well as behavioural plasticity [94,95]. Small RNAs profiling in bee larval food also revealed that worker jelly displays an increased miRNA complexity and abundance compared with royal jelly. These results indicate a potential important role for miRNAs in the regulation of gene expression in caste polyphenism [96]. Similar patterns of caste-specific expression of miRNAs were also reported in the bumblebee *Bombus terrestris* [97].

Other classes of small RNAs, including siRNAs and piRNAs have not received much attention yet. Long non-coding RNAs (lncRNAs) are also important regulators of gene expression that can act transcriptionally or post-transcriptionally [98]. For instance, honeybees display two lncRNAs (lncov1 and lncov2) with a potential role in regulating the size of the worker ovaries [99]. Altogether, non-coding RNAs profiling revealed caste-specific expression in various species, but very little is known on their exact functional role in the context of polyphenism.

### 4.5. Conclusions

Polyphenism appears to be an extraordinarily efficient way for organisms to cope with the various and predictable fluctuations of their environment by producing better-adapted phenotypes. This relies on the genome ability to switch rapidly from a developmental program to another. Insects display a remarkable variety of polyphenisms, ranging from seasonal adaptations, the ability to optimize their dispersal strategies and the division of labour within a colony. In all cases, epigenetic mechanisms are key actors of this genome expression plasticity (summarized in Table 1); however, their precise role in the context of polyphenisms remains to be elucidated, i.e., if they trigger the changes, or simply facilitate or maintain them. Caste polyphenism has thus far received the main attention and especially in well-studied organisms, such as ants and bees, several studies pointing out the role of DNA methylation and alternative splicing in the establishment of caste-specific transcriptional programs underlying alternative phenotypes. The investigation of genome-wide chromatin dynamics also revealed that specific combinations of histone PTMs were strongly associated with the expression of distinct sets of genes, thereby with caste identity. In the case of seasonal and dispersal polyphenism, evidence for the functional role of epigenetic mechanisms in the regulation of alternative phenotypes are so far limited, with the notable exception of the characterization of the role of DNA methylation during the phase polyphenism in locusts. Various transcriptomic studies comparing alternative phenotype production identified many putative candidate genes that could play a role in neuro-endocrine transduction and epigenetic regulation. Nevertheless, their causality is not demonstrated yet, in part because these non-model organisms often suffer from the absence of efficient functional validation tools. DNA methylation analyses is relatively easy to develop; however, the challenge in the near future will rely on setting up molecular techniques (such as ChIP-seq or ATAC-seq for example) to implement genome-wide chromatin dynamics analysis, allowing to reveal the epigenomic identity of alternative phenotypes. In addition, these studies will have to be investigated at the critical steps (and potentially in specific tissues) when (and where) developmental switches occur rather than on final phenotypes. This will then allow for identifying the key factors that are responsive to the environmental signals and initiate the cascades of gene expression changes that finally contribute to the establishment of the different phenotypes.

## Figures and Tables

**Figure 1 insects-12-00649-f001:**
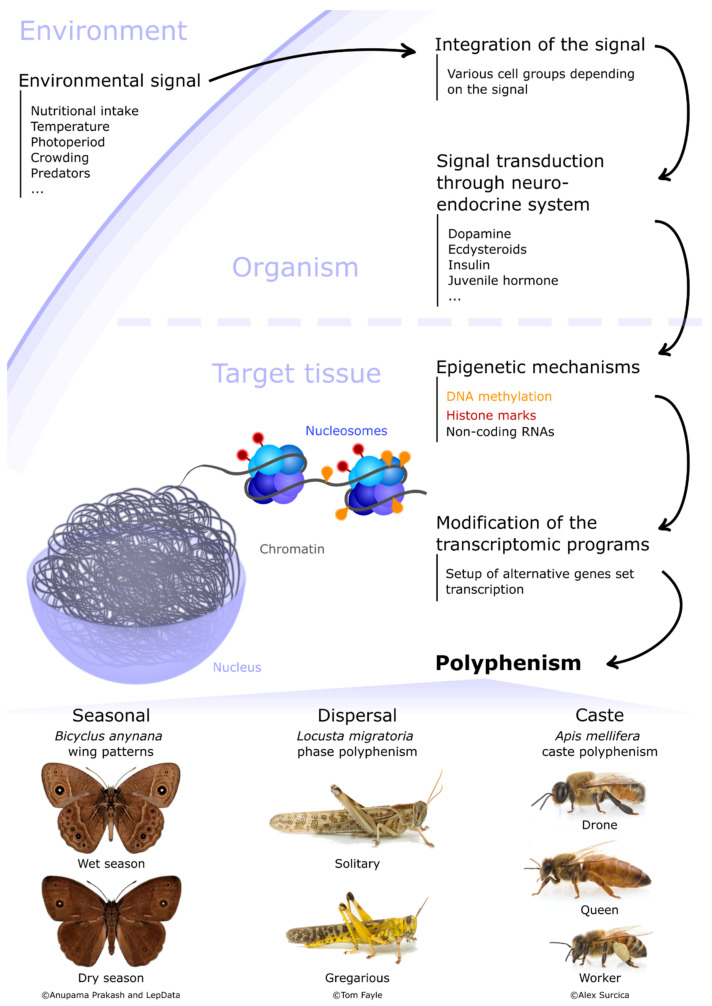
Insect polyphenism regulation framework. Environmental signal(s) are first perceived and integrated by specific groups of cells. This signal is then transduced to the target tissues/cells by the neuro-endocrine system, often relying on multiple hormones and neuropeptides. Epigenetic mechanisms mainly involving chromatin regulation and DNA methylation are then activated in specific cells of the target tissue(s). These mechanisms then modulate gene expression, ending up with the setup of alternative transcriptional programs underlying the production of discrete phenotypes. Examples of discrete phenotypes for three types of polyphenism are illustrated here: (i) the seasonal polyphenism of *Bicyclus anynana* with wet (**top**) and dry (**bottom**) season morphs (credits: Anupama Prakash, University of Singapore and LepData, 2018); (ii) the phase polyphenism of *Locusta migratoria,* with a solitary (**top**) and a gregarious (**bottom**) male (credits: Tom Fayle, University of Cambridge); (iii) the caste polyphenism of *Apis mellifera* with drone (**top**), queen (**middle**) and worker (**bottom**) morphs (credits: Alex Surcica, Digital Museum of Natural History).

**Table 1 insects-12-00649-t001:** Summary of the epigenetic mechanisms involved in the regulation of environmentally-induced polyphenism in insects.

	Insect Species	Phenotypes	Environmental Signal(s)	Neuro-Endocrine System	Epigenetic Mechanism Studied				Key References
					Chromatin Regulation	DNA Methylation	sRNAs	lncRNAs	
Seasonal polyphenism	*Bicyclus anynana* (Nymphalida)	Wings spots patterns	Temperature	Ecdysteroids	-	-	-	-	-
	*Acyrthosiphon pisum* *Myzus persicae* (Aphidida)	Clonal Sexual	Photoperiod	Dopamine, Insulin, Juvenile hormone	Morph-specific open chromatin profile of the X chromosome	Morph-specific DNA methylation profile of the X chromosome	-	-	[44,45]
Dispersal polyphenism	*Schistocerca gregaria* *Locusta migratoria* (Acrididae)	Solitary Gregarious	Visual, olfactory and physical contacts	Corazonin, Dopamine, Juvenile hormone, Serotonin	H3 phosphorylation in gregarious locusts brains	Differentially methylated genes between phases	Phase-specific sRNAs profiles	-	[47,58,59,60,61]
	*Acyrthosiphon pisum* (Aphididae)	Winged Wingless	Crowding, Predators, Food quality	Ecdysone, Juvenile hormone	-	-	-	-	[65]
Caste polyphenism	*Apis mellifera* (Apidae)	Queen Worker	Larval diet, Genetic factors	Insulin, Juvenile hormone	HDACi in royal jelly Caste-specific patterns of H3K4me3, H3K27ac, H3K36me3	Differentially methylated genes between castes. Correlation between methylation and alternative splicing	Caste-specific miRNAs profiles	Caste specific lncRNAs (lncov1, lncov2)	[75,79,80,81,82,85,86,87,93,94,95,96,99]
	*Camponotus floridanus* *Harpegnathos saltator* (Formicidae)	Queen Male Worker (Gamergate)	Larval diet Pheromones Genetic factors	Corazonin	H3K27ac strongly associated with caste identity Differential binding of CBP	Differentially methylated genes between castes. Correlation between methylation and alternative splicing	Caste-specific miRNAs profiles	-	[83,89,90,92]
	*Zootermopsis nevadensis* (Archotermopsidae)	Queen Worker	Larval diet	Juvenile hormone	-	Differentially methylated genes between castes. Correlation between methylation and alternative splicing	-	-	[84]

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
