# Peer review of "Contribution of Epigenetic Mechanisms in the Regulation of Environmentally-Induced Polyphenism in Insects"

_insects, 2021, doi:10.3390/insects12070649_

Round 1
Reviewer 1 Report
The review by Richard et al. presents a useful synopsis of observations linking environmentally-induced polyphenisms to alterations in epigenetic marks, focusing on some insect species. There is some value in synthesizing these data, however there are some concerns that should be addressed.
- Introduction section. Since the discovery of 5hmC, 5fC, 5caC and 6mA, DNA cytosine methylation is no longer the only epigenetic mechanism involving covalent modification of DNA. Moreover, exchange of histone variants and three-dimensional architecture of chromatin are also involved in the epigenetic control of gene expression. For example, in aphids structurally distinct chromosomes form during parthenogenesis and sexual reproduction, often in response to environmental changes. The manuscript would be significantly strengthened if these points were comprehensively addressed in the Introduction section and/or in the subsequent paragraphs.
- Additional examples of epigenetically-regulated polyphenisms in insects should be included. Among others, a pertinent study that deserves to be discussed pertains the increase in mandible size in male broad-horned flour beetles, which depends upon nutrient availability via histone acetylation [PNAS USA 113, 15042–15047].
- The authors recognized that literature adequately justifying the involvement of epigenetic mechanisms for wing patterning polyphenism in butterflies and wing polyphenism in aphids is missing so far (lines 139-142 and 307-308, respectively). Being the contribution of epigenetic mechanisms the focus of this review, both the mentioned paragraphs should be removed.
- The authors write “They found that the chromatin of the single X chromosome of males was more accessible than the chromatin of the two Xs of females” (lines 184-185). This could most probably due to the intrinsic dosage compensation program occurring in this organism rather than to environmental influence. To avoid reader confusion, the authors should clarify this point.
- I suggest rewriting and expanding the Conclusion section, which seems quite redundant with the previous paragraphs. In particular, in my view the authors should highlight that it is not clear whether epigenetic mechanisms cause, facilitate, or simply maintain polyphenisms once transcriptional differences have been triggered by environmental stimuli.
- The whole manuscript needs to be more carefully edited. I did not note all examples, but some of them are as follows: “epigenetic” (line 87) should be “epigenetics”, “histone" (line 260) should be “histones”, “PTHMs” (line 406) should be “PTMs”, and more. Furthermore, it is not clear to me what the authors mean by “stipulation” (line 140).
Author Response
Reviewer 1
The review by Richard et al. presents a useful synopsis of observations linking environmentally-induced polyphenisms to alterations in epigenetic marks, focusing on some insect species. There is some value in synthesizing these data, however there are some concerns that should be addressed.
1 - Introduction section. Since the discovery of 5hmC, 5fC, 5caC and 6mA, DNA cytosine methylation is no longer the only epigenetic mechanism involving covalent modification of DNA. Moreover, exchange of histone variants and three-dimensional architecture of chromatin are also involved in the epigenetic control of gene expression. For example, in aphids structurally distinct chromosomes form during parthenogenesis and sexual reproduction, often in response to environmental changes. The manuscript would be significantly strengthened if these points were comprehensively addressed in the Introduction section and/or in the subsequent paragraphs.
We do agree that these recently discovered modifications of DNA might be important regulators of polyphenism. As well, histone variants but also long-range interactions in the nucleus revealed by HiC analyses might be key contributors. We added sentences in the introduction to be more exhaustive regarding the variety of epigenetic mechanisms. Nevertheless, there is – to our knowledge – no evidence of a role of these mechanisms in the control of insect polyphenisms. Regarding the assumption that “In aphids structurally distinct chromosomes form during parthenogenesis and sexual reproduction”, I am not sure this is accurate. In the case of the pea aphid, parthenogenetic females and sexual females display 3 pairs of autosomes and 2 X chromosomes, while males display 3 pairs of autosomes and one X chromosome. The chromatin structure of the X seems to differ between males and females (whether sexual or parthenogenetic). However, since males and females also differ by the number of X copies, we cannot disentangle whether sexual dimorphism originates from the number of X copies, or by the chromatin structure of the X. Anyway, the example of male and females aphids cannot be considered a true polyphenism since their karyotypes differ.
2 - Additional examples of epigenetically-regulated polyphenisms in insects should be included. Among others, a pertinent study that deserves to be discussed pertains the increase in mandible size in male broad-horned flour beetles, which depends upon nutrient availability via histone acetylation [PNAS USA 113, 15042–15047].
The study on broad-horned beetles documents the role of HDACs in the variation of the size of the mandible. This continuous variation in mandible size cannot be considered as a polyphenism (which imply discrete morphs), but is a good example of phenotypic plasticity. As our review is about polyphenims, we have not included it in the reviewed examples. We thoroughly checked the literature and did not find other insects’ polyphenisms including documented epigenetic mechanisms, which is the focus of this review.
3 - The authors recognized that literature adequately justifying the involvement of epigenetic mechanisms for wing patterning polyphenism in butterflies and wing polyphenism in aphids is missing so far (lines 139-142 and 307-308, respectively). Being the contribution of epigenetic mechanisms the focus of this review, both the mentioned paragraphs should be removed.
We agree that the contribution of epigenetic mechanisms for wing patterning polyphenism in butterflies and wing polyphenism in aphids is currently not demonstrated. Nevertheless, transcriptomic analyses suggest the involvement of epigenetic regulations (especially in the case of wing polyphenism) which is a first hint towards the investigation of these mechanisms. The other reviewers also found these sections informative, that is why we would prefer to keep it for publication.
4 - The authors write “They found that the chromatin of the single X chromosome of males was more accessible than the chromatin of the two Xs of females” (lines 184-185). This could most probably due to the intrinsic dosage compensation program occurring in this organism rather than to environmental influence. To avoid reader confusion, the authors should clarify this point.
That’s a fair point. The production of sexual morphs is the end-point of reproductive polyphenism, and the enhanced chromatin accessibility of the X in males is likely to be due to a dosage compensation mechanism. Nevertheless, this study represents the very first example demonstrating the existence of epigenome (-wide) differences between plastic morphs in aphids. Histone patterns comparisons in ants are also focused on “finished” morphs, which is very comparable. We clarified this point in the text.
5- I suggest rewriting and expanding the Conclusion section, which seems quite redundant with the previous paragraphs. In particular, in my view the authors should highlight that it is not clear whether epigenetic mechanisms cause, facilitate, or simply maintain polyphenisms once transcriptional differences have been triggered by environmental stimuli.
Thank you for this suggestion. We have modified the conclusion accordingly.
6 - The whole manuscript needs to be more carefully edited. I did not note all examples, but some of them are as follows: “epigenetic” (line 87) should be “epigenetics”, “histone" (line 260) should be “histones”, “PTHMs” (line 406) should be “PTMs”, and more. Furthermore, it is not clear to me what the authors mean by “stipulation” (line 140).
Thank you. These mistakes have been corrected. The manuscript has been carefully edited.
Reviewer 2 Report
Dear colleagues,
Thank you for giving me the opportunity to read the review from Richard, Jaquiéry and Le Trionnaire. In my opinion, this review has span over all th epigenetic mechanisms that contribute to the regulation of environmentally-induced polyphenism in insects. This article sounds and we can see that the authors have a good knowledge in the field, both from the Ecology and the Molecular aspects. As such, I will have only (regular) miscellaneous typos and small mistakes to deal with.
L 46: Phenotype --> PhenotypIC
Keep coherent decision whether you want to have a first upper case character after a period. E.g. L105 piRNAs L106 LncRNAs
L136: of the dry morph --> of the dry season morph
L169: what else could it be? (the circadian clock might be a component of the photoperiodic clock)
Acronyms are not necessary when they are just used a very few times. If so, keep the full name (e.g. L 429). Especially when used at distant parts of the manuscript (e.g. juvenile hormone L162, L224, L280, L353). In this respect, either say what JH means in the caption of the table, or better, write in full.
L224: differencing the brown and solitary to the gregarious harboring vivid colors (otherwise brown AND vivid L225)
L257: please refine the interpretation.
L339: short definition of hybridogenesis could be helpful
L380: alk, a receptor tyrosine kinase, ...
L406: a 96-hour-old (hyphens and no plural form)
L450, L466: do not cut the word here: miRNA
L233: what species is the desert locus?
L241: what is a "takeout" protein?
L250: recheck first use of a species in full, along with its vernacular name, then keep first letter of the genus afterward. (L223, L233 L250 L251 etc)
L442: CBP?
L467: in regulating the size of the worker ovaries
L494: again, watch the plural form in complexe sentences (dynamicS analyseS)
Figure 1.
Please write the species name in full. Add the peculiarity of these morphs (e.g. wet season morph, dry season morph, solitary, gregarious, drone, queen, worker)
Caption: remove articles, "a" and "the", e.g. : with wet and dry season morphS (plural is missing), with drone, queen and worker phorphs.
Reference 2, 3rd author is Gerdien de Jong (de Jong G as in reference 14 de Jong MA)
There are several works from Soojin Yi's (U. of Georgia, https://yilab.gatech.edu/publications/) that should very much interest the authors and are missing among the fair amount of references cited.
Ref. 58 gap is missing between "in" and "Myrmica"
Ref 59: twice Social Insects
References, month is not provided everywhere (see ref 82, written in French).
Species names should be written in italic.
Article titles sometimes have an upper case character for each words, some others don't
Author Response
Reviewer 2
Dear colleagues,
Thank you for giving me the opportunity to read the review from Richard, Jaquiéry and Le Trionnaire. In my opinion, this review has span over all the epigenetic mechanisms that contribute to the regulation of environmentally-induced polyphenism in insects. This article sounds and we can see that the authors have a good knowledge in the field, both from the Ecology and the Molecular aspects.
As such, I will have only (regular) miscellaneous typos and small mistakes to deal with.
All typos were corrected, and specific points/questions are addressed here.
L 46: Phenotype --> PhenotypIC
Keep coherent decision whether you want to have a first upper case character after a period. E.g. L105 piRNAs L106 LncRNAs
L136: of the dry morph --> of the dry season morph
L169: what else could it be? (the circadian clock might be a component of the photoperiodic clock)
The question of time measurement in insect photoperiodism is a heavily debated topic (see Saunders, 2005 – “Erwin Bünning and Tony Lees, two giants of chronobiology, and the problem of time measurement in insect photoperiodism”). Different models have been proposed including one where the photoperiodic clock might be independent from the circadian clock (“The non-circadian hourglass-like timer”) and another one where the circadian clock serves as an input of the photoperiodic clock (“A circadian oscillatory clock”). So far, it is still unclear.
Acronyms are not necessary when they are just used a very few times. If so, keep the full name (e.g. L 429). Especially when used at distant parts of the manuscript (e.g. juvenile hormone L162, L224, L280, L353). In this respect, either say what JH means in the caption of the table, or better, write in full.
“JH” has been replaced by “juvenile hormone” throughout the manuscript.
L224: differencing the brown and solitary to the gregarious harboring vivid colors (otherwise brown AND vivid L225)
L257: please refine the interpretation.
We added a sentence.
L339: short definition of hybridogenesis could be helpful
We added a sentence.
L380: alk, a receptor tyrosine kinase, ...
We added a sentence.
L406: a 96-hour-old (hyphens and no plural form)
Corrected.
L450, L466: do not cut the word here: miRNA ???
This is due to text re-formatting by Insects journal. We will be careful when correcting the proofs.
L233: what species is the desert locus?
Already specified a few lines above, the desert locust is Schistocerca gregaria.
L241: what is a "takeout" protein?
We added a sentence.
L250: recheck first use of a species in full, along with its vernacular name, then keep first letter of the genus afterward. (L223, L233 L250 L251 etc)
L442: CBP?
CBP stands for CRB Binding Protein. The definition of this acronym is provided in the text.
L467: in regulating the size of the worker ovaries
Corrected.
L494: again, watch the plural form in complexe sentences (dynamicS analyseS)
Corrected.
Figure 1.
Please write the species name in full. Add the peculiarity of these morphs (e.g. wet season morph, dry season morph, solitary, gregarious, drone, queen, worker)
Caption: remove articles, "a" and "the", e.g. : with wet and dry season morphS (plural is missing), with drone, queen and worker phorphs.
Thank you. The figure has been edited accordingly.
Reference 2, 3rd author is Gerdien de Jong (de Jong G as in reference 14 de Jong MA)
There are several works from Soojin Yi's (U. of Georgia, https://yilab.gatech.edu/publications/) that should very much interest the authors and are missing among the fair amount of references cited.References from Soojin Yi’s lab are indeed of high interest for the field. We thus added two references in relation with DNA methylation in insects, notably in relation with Apis mellifera caste polyphenisms.
Ref. 58 gap is missing between "in" and "Myrmica"Ref 59: twice Social InsectsReferences, month is not provided everywhere (see ref 82, written in French). Species names should be written in italic.
Article titles sometimes have an upper case character for each words, some others don't
Thank you. We corrected all these errors. The reference section has now been carefully checked.
Reviewer 3 Report
The manuscript by Richard at al. presents a review of epigenetic control over trait polyphenism in insects. Overall, the MS is a well-written and comprehensive treatment of a developing field. I have no major issues (only minor ones below) with the possible exception that the categories of polyphenisms are somewhat fuzzy. For example, aphids are considered a reproductive polyphenism as they switch from parthenogenic to sexual as cued by day length. But this is also a dispersal and a host preference polyphenism (and a seasonal one too). I also wondered if these categories were incomplete or perhaps poorly defined. What about seasonal host switching and defense polyphenisms (there is a damselfly example for the latter)? If the authors are ignoring examples/categories where no epigenetic work has been done then they should say as much. Otherwise, perhaps just another sentence or two recognizing fuzzy borders and potential additional categories would suffice.
This MS is rich in detail -- I was able to follow it but it is quite dense. I do not feel equipped to say whether all the details were precisely correct and complete. That said, the authors appear to command the topic so I tend to believe that their treatment of the mechanisms is a good one.
Minor issues and edits:
L12: "summarizes the knowledge" Remove "the"
L17: "The nature of the cues" Prefer just "cues"
L26: "Combinatorial patterns of histone PTMs provide phenotype-specific epigenomic landscape " Word missing here.
L: "RNAs profiles" "RNA profiles" (no s)
L30: "knowledge on" "knowledge OF"
L39: "aptitude" prefer "ability"
L101: "genomic context" Perhaps consider elaborating on this "genomic context" idea.
L105: "for piwi-interacting RNA" Please explain what these are.
Fig. 1 No "e" in "insulin" (also in Table)
Fig. 1: "for piwi-interacting RNA" Nice figure -- just make sure it is consistent with the text. i.e., you say there are three mechanisms of phenotype modification but then list four.
L144: "aphids...are crop pests" I would say that aphids are a large family of hemipterous insects characterized by complex life cycles on diverse and often multiple hosts -- some of these are crop pests.
L156: As stated above, aphid reproductives also have wings and very often disperse to unrelated host plants where they feed and overwinter. You're underselling the phenotypic changes here which are fascinating and complex. This does add some complexity to your classification scheme though as this is now a dispersal and host preference polyphenism too, but that's the reality. I see that you mention this later but it may warrant more consderation here.
L140: " distal-less gene in the stipulation of a focal" is "stipulation" the right word here?
L172: "epigenetic modifications that trigger these early transcriptomic changes." This might be my ignorance but is it 100% known that plasticity must have an epigenetic component? Could the environment not trigger gene expression more directly somehow?
L214: "Contrastingly" In contrast?
L335: "pheromone" pheromones (plural)
L335: "orientate" prefer "alter" or "influence"
L361: "Contrastingly" Not sure if this is a word, but "In contrast" is definitely more common.
L369: "550" Write out # at the beginning of a sentence.
Author Response
Reviewer 3
The manuscript by Richard at al. presents a review of epigenetic control over trait polyphenism in insects. Overall, the MS is a well-written and comprehensive treatment of a developing field. I have no major issues (only minor ones below) with the possible exception that the categories of polyphenisms are somewhat fuzzy. For example, aphids are considered a reproductive polyphenism as they switch from parthenogenic to sexual as cued by day length. But this is also a dispersal and a host preference polyphenism (and a seasonal one too). I also wondered if these categoies were incomplete or perhaps poorly defined. What about seasonal host switching and defense polyphenisms (there is a damselfly example for the latter)? If the authors are ignoring examples/categories where no epigenetic work has been done then they should say as much. Otherwise, perhaps just another sentence or two recognizing fuzzy borders and potential additional categories would suffice.
We do agree with the referee comment. For example in aphids, we focused on monoecious species that do not change of host plant during their life cycle. Heteroecious species indeed produce intermediate morphs (between clonal and sexual morphs), such as the gynoparae (the sexual female producer), which is actually the morph that switches from summer host plant to winter host plant and display at this occasion a nutrition polyphenism. This interplay between a reproductive and a nutritional polyphenism is indeed of interest but suffers from the absence of any molecular data, explaining why it is not included in the review. Regarding defensive polyphenism, this is also of interest but we did not focus on this example considering the lack of epigenetic studies. Nevertheless, we do agree that for example in aphids dispersal polyphenism can also be considered somehow as a defensive polyphenism (when induced by predators). We used this example in the text to point out these fuzzy borders.
This MS is rich in detail -- I was able to follow it but it is quite dense. I do not feel equipped to say whether all the details were precisely correct and complete. That said, the authors appear to command the topic so I tend to believe that their treatment of the mechanisms is a good one.
Minor issues and edits:
All typos were corrected, and specific points/questions are addressed here.
L12: "summarizes the knowledge" Remove "the"
L17: "The nature of the cues" Prefer just "cues"
L26: "Combinatorial patterns of histone PTMs provide phenotype-specific epigenomic landscape " Word missing here.
L: "RNAs profiles" "RNA profiles" (no s)
L30: "knowledge on" "knowledge OF"
L39: "aptitude" prefer "ability"
L101: "genomic context" Perhaps consider elaborating on this "genomic context" idea.
L105: "for piwi-interacting RNA" Please explain what these are.
Fig. 1 No "e" in "insulin" (also in Table)
Fig. 1: "for piwi-interacting RNA" Nice figure -- just make sure it is consistent with the text. i.e., you say there are three mechanisms of phenotype modification but then list four.
L144: "aphids...are crop pests" I would say that aphids are a large family of hemipterous insects characterized by complex life cycles on diverse and often multiple hosts -- some of these are crop pests.
We added a sentence.
L156: As stated above, aphid reproductives also have wings and very often disperse to unrelated host plants where they feed and overwinter. You're underselling the phenotypic changes here which are fascinating and complex. This does add some complexity to your classification scheme though as this is now a dispersal and host preference polyphenism too, but that's the reality. I see that you mention this later but it may warrant more consderation here.
As said above, we clarified this point in the text.
L140: " distal-less gene in the stipulation of a focal" is "stipulation" the right word here?
This is indeed the word used in the original paper.
L172: "epigenetic modifications that trigger these early transcriptomic changes." This might be my ignorance but is it 100% known that plasticity must have an epigenetic component? Could the environment not trigger gene expression more directly somehow?
It is not sure for 100%, but such rapid changes in gene expression that provide such discrete phenotypic changes must be triggered by genome-wide modifications. Epigenetic and transcriptional regulation thus appear as logical candidates, but Post-Transcriptional Gene Silencing (PTGS) and Translational control mechanisms might also play role. But there is so far no evidence.
L214: "Contrastingly" In contrast?
L335: "pheromone" pheromones (plural)
L335: "orientate" prefer "alter" or "influence"
L361: "Contrastingly" Not sure if this is a word, but "In contrast" is definitely more common.
L369: "550" Write out # at the beginning of a sentence.
Round 2
Reviewer 1 Report
The authors have addressed my concerns and I believe that the paper is ready for publication.